# A Role for the Microbiota in the Immune Phenotype Alteration Associated with the Induction of Disease Tolerance and Persistent Asymptomatic Infection of *Salmonella* in the Chicken

**DOI:** 10.3390/microorganisms8121879

**Published:** 2020-11-27

**Authors:** Annah Lee, Cristiano Bortoluzzi, Rachel Pilla, Michael H. Kogut

**Affiliations:** 1Department of Poultry Science, Texas A&M University, College Station, TX 77843, USA; annahlee89@gmail.com (A.L.); bortoluzzi.c@gmail.com (C.B.); 2Department of Small Animal Clinical Sciences, Texas A&M University, College Station, TX 77843, USA; rpilla@cvm.tamu.edu; 3Food and Feed Safety Research Unit, United States Department of Agriculture/Agriculture Research Service, College Station, TX 77845, USA

**Keywords:** *Salmonella enteritidis*, gut microbiota, antibiotics, immune phenotype

## Abstract

Previous studies have shown a tissue immune phenotype-altering event occurring on days 2 and 4 in the ceca post-*Salmonella* challenge. To evaluate the involvement of the cecal microbiota in the phenotype reprogramming, we hypothesized that the addition of subtherapeutic bacitracin (BMD) will affect the cecal microbiota. Therefore, the objective of this study was to determine if the antibiotic-mediated changes in the microbiota composition influenced the immune phenotype induced by *Salmonella enteritidis* infection of the chicken cecum. A total of 112 fertile eggs were obtained for each experiment, repeated for a total of three separate times. The ceca and cecal contents were collected on days 2 and 4 post-infection for mRNA expression TaqMan assay and 16S rRNA gene microbiota sequencing. The results demonstrate the effects of bacitracin on cecal composition and its interaction with *Salmonella enteritidis* in young chicks. There is a preliminary indication of phenotype change in the *Salmonella*-challenged group provided subtherapeutic BMD due to the shifting cecal microbiota and cecal immune response, indicating the addition of bacitracin during infection altered the cecal phenotype. These data demonstrate the potential involvement of the microbiota in reprogramming immune phenotype (disease resistance to disease tolerance) induced by *Salmonella* in the chicken cecum.

## 1. Introduction

Foodborne illnesses cause health and economic burden in the United States annually, affecting 48 million people every year. One of the major causes of human gastroenteritis is *Salmonella enterica* Enteritidis (*S.* Enteritidis) due to infected poultry products, accounting for 40–60% of all reported cases [1]. Although improved control measures have been implemented, *S. enterica* still continues to present an issue every year in US livestock [2]. One of the many reasons is due to the chickens’ ability to co-exist with *Salmonella* without showing any outward clinical signs of distress, also referred to as disease tolerance, and increased antimicrobial resistance due to overuse of antibiotic growth promoters (AGPs) [3]. Although the mechanism of how AGPs improve animal performance is still unclear, Diaz Carrasco et al. [4] speculate it is through intestinal microbiota modulation, specifically in the ceca for the chicken, whether it is dietary related or pathogen related. Studies need to focus on identifying the mechanism of how broad-spectrum antibiotics work in the host gut towards improving the overall health of food production animals.

In the poultry industry, bacitracin methylene disalicylate (BMD) is used as an AGP to improve poultry health and growth development by reducing inflammatory markers [5]. The mechanism of action for BMD is targeting Gram-positive bacteria, which are usually associated with poorer animal performance and growth [6]. Furthermore, BMD promotes growth, proliferation, or intactness of beneficial and resident bacterial gut communities to confer protection against pathogens, such as *Salmonella enterica* [7]. What is important to note is that to promote an optimal immune system, the host needs a balanced pro-inflammatory and anti-inflammatory response. The suppression of pro-inflammatory responses does not always reap benefits to the host immunity. In fact, the initial immune response may be less effective while allowing pathogens to proliferate on the opportunity of imbalance [8]. However, most of these studies have been performed in vitro, which concluded in variable results of AGP and its effects on phagocyte or immune cell functions [8]. Therefore, focusing on the gut microbiota with subtherapeutic BMD inclusion may provide insights on improved mechanisms and health management strategies. Increasing studies are showing the importance of the gut microbiota and its role in digestion, host immunity, immune protection, and protection against pathogen colonization [9]. However, antibiotic usage in US poultry is facing scrutiny due to antibiotic resistance and increased consumer awareness [10]. Currently, many European countries have banned the usage of AGP in animal feed, even at subtherapeutic levels; in the US, subtherapeutic levels of AGP must now be provided under veterinary authorization [11].

Even before hatch, the chick can be infected with *S.* Enteritidis via vertical transmission [12]. Commercial broiler chicks are placed on used litter at hatch, which raises the risk of the naïve gut microbiota of neonatal chicks to be colonized by pathogenic *Salmonella* and take advantage of this susceptible environment [2,13,14,15]. The ceca contain the most diverse bacterial populations in the gastrointestinal tract (GIT): phyla *Bacteroides*, *Proteobacteria* and *Firmicutes* are the most dominantly observed [4,9,16]. Many of the taxa present in the chicken GIT are of human relevance, including *Salmonella* and certain *Campylobacter* species [16]. Recent findings have speculated a phenomenon occurring in the chicken ceca with *S.* Enteritidis infection, resulting in three distinct immune-metabolic phases: disease resistance (4–48 h post-infection), disease tolerance (3–4 days post-infection), and homeostasis (5–28 days post-infection) [17,18]. The study of the symbiosis between the metabolic and immunological (immunometabolic) changes occurring in the gut microbiota during *Salmonella* infection is a growing field, especially in the chicken. Once *Salmonella* enters the intestine, the interaction with the mucosal surface is crucial for successful attachment. Inflammatory responses can occur in as little as few hours after oral ingestion; however, the damage to the host does not occur in the chicken. Instead, a state of persistence occurs, allowing for minimized host defense action. While this is not harmful for the health of the chicken, it is not ideal downstream for human consumption. This phenotypic alteration event has been theorized as a survival mechanism of *Salmonella* in poultry to minimize host defenses [3].

Therefore, the objective of this study was to evaluate the effects of the broiler cecal microbiota and immunity induced by *Salmonella* infection with the addition of subtherapeutic level of BMD. The hypothesis is that tissue (cecal) phenotype will be altered due to the addition of subtherapeutic BMD during *Salmonella* infection. The results from this study will provide crucial perspectives on how broad range antibiotics act on the ceca, especially during a *Salmonella* infection, and potentially lead to better improvements on poultry intestinal health.

## 2. Materials and Methods

### 2.1. Experimental Animals, Housing, and Treatments

All experiments conducted were in accordance with the guidelines set by the United States Department of Agriculture Animal Care and Use Committee (IACUC #2019-003). By-product Cobb 500 broiler eggs were obtained from a commercial hatchery. Before incubation, the randomly selected eggs were swabbed for *Salmonella* detection on the shell surface. *Salmonella* were not detected on any of the tested surfaces. The fertile eggs were incubated (GQF Manufacturing Company, Savannah, GA, USA; Jamesway Incubator Company, Inc., Cambridge, ON, Canada; or Petersime Incubator Co., Gettysburg, PA, USA) and maintained at wet and dry bulb temperatures of 32 and 37 °C, presented as relative humidity. After 10 days of incubation, the eggs were candled and non-fertile or non-viable eggs were discarded. The viable eggs were returned to the incubator until day 18 when they were transferred to hatchers (Humidaire Incubator Company, New Madison, OH, USA) and maintained under the same temperature and humidity conditions until hatch. At hatch (*n* = 112), the chicks were randomly distributed into one of four groups (28 chicks/group): (T1) non-infected birds fed a broiler starter diet alone; (T2) *S.* Enteritidis-challenged birds fed a broiler starter diet alone; (T3) non-infected birds fed a broiler starter diet containing BMD (Zoetis Inc., Parsippany, NJ, USA) at the inclusion rate of 50 g/ton; (T4) *S.* Enteritidis-challenged birds fed a broiler starter diet containing BMD at the inclusion rate of 50 g/ton.

At day of hatch, the chicks were randomly distributed into each treatment group in pens with fresh pine shavings, water, and starter diet ad libitum. The chicks were maintained in biosafety level-2 (BSL-2) isolation units under 96 h light and then switched over to an 18 h light:6 h dark cycle until the end of the study. The temperature in the room was held between 90 to 95 °F (temperature decreasing to 90 °F by day 10). Birds were monitored daily for the entire experimental period. This study was repeated three times under identical parameters.

### 2.2. Bacteria Preparation

Upon hatching, all chicks were orally challenged with 10^6–8^ CFU/0.5 mL *S.* Enteritidis or mock challenge with 0.5 mL sterile 1xPBS. The *Salmonella enterica* serovar Enteritidis isolate was obtained from the National Veterinary Services Laboratory (Ames, IA, USA), selected for resistance to novobiocin (25 ug/mL) and nalidixic acid (20 ug/mL) in tryptic soy broth (Difco Laboratories, Sparks, MD, USA). Viable cell concentration of challenge were determined on XLT4 agar plates with XLT4 supplement (Difco Laboratories, Sparks, MD, USA), nalidixic acid and novobiocin (XLT-NN).

### 2.3. Bacterial Detection

The cecal contents (0.25 g/bird) were collected, serially diluted to 1:100, 1:1000, 1:10,000, 1:100,000, and spread onto XLT4 for *Salmonella* detection. These plates were incubated at 37 °C for 24 h. For the enrichment, 100 uL of the cecal contents were pre-enriched in Rappaport–Vassiliadis broth for 24 h at 37 °C.

### 2.4. Sample Collection and Processing

On each necropsy day (day 2 and 4 post-infection), 14 birds/group were selected by random and euthanized via cervical dislocation. The cecal contents were collected for bacterial detection and microbiota analysis. The ceca were stored in RNALater for gene expression studies. This trial was repeated two more times for three total separate experiments.

### 2.5. Microbiota Sequencing

For the microbiota studies, the remaining contents from each ceca (ranging from 300–500 mg per cecal sample) were submitted to a core sequencing facility at the University of Arkansas, Fayetteville (*n* = 14/treatment/day/experiment; 336 cecal contents submitted total). DNA was extracted from the ceca samples taken at the indicated time points (day 2 and 4) using the Qiagen Qiamp Fast DNA Stool Mini Kit (Qiagen, Hilden, Germany). The DNA purity was assessed, and all of the DNA samples were diluted to 10 ng/mL. The paired-end sequencing libraries were prepped by targeting the hypervariable region 4 of the 16S ribosomal RNA with PCR primers containing the linker and adapter sequence. The sequencing libraries were assessed for qualitative and quantitative homogeneity, and then sequenced using the Illumina MiSeq platform as previously described [19].

### 2.6. Microbiota Bioinformatic Analysis

Data sequences were uploaded onto BaseSpace (www.basespace.illumina.com) (Illumina, San Diego, CA, USA) to determine sequence run quality and run completion. De-multiplexed data were downloaded locally and uploaded onto QIIME2-2018.8. The sequences were demultiplexed and the amplicon variant sequence (ASV) table was created using DADA2 [20]. The representative sequences were then classified using VSEARCH, and ASVs were defined as sequences with at least 97% similarity with Greengenes v.13.8 database [21]. Sequences that were unassigned or identified as cyanobacteria, chloroplast, or mitochondria were removed. The ASV table was then rarefied at 6400 reads for even depth of analysis. Alpha diversity metrics calculated were Shannon’s diversity index, observed OTUs (operational taxonomic units), and Chao1. Beta diversity metrics that were estimated with unweighted and weighted UniFrac distance and visualized using principle coordinate analysis (PCoA). Nonparametric test ANOSIM (analysis of similarities) was used to compare the similarity between the bacterial composition of each treatment to the baseline control (T1) using UniFrac.

### 2.7. Real-Time Quantitative RT-PCR

The immune genotyping portion was quantitated by mRNA expression, specifically through a TaqMan based assay adapted from Eldaghayes et al. [22]. The ceca that were previously stored in RNALater were cut longitudinally to expose the lumen, and any remaining fecal matter was gently removed with forceps. The total RNA was extracted using Qiagen RNeasy^®^ Plus kits (Germantown, MD, USA), and RNA quality was evaluated with a NanoDrop^TM^ 2000 Spectrophotometer (Thermo Fisher Scientific, Waltham, MA, USA).

Cytokine mRNA expression levels were determined using RT-PCR with 28S as the reference gene. The RNAs were stored in −80 °C until RT-PCR plate setup. The cytokines IL-1β, IL-6, IL-10, IFN-γ, and TNF-α were quantified utilizing the Eldaghayes et al. method [22]. Primer and probe sequences (Table 1) for amplification have been described previously by Kogut et al. [23] and Kaiser et al. [24]. The plates were run in the Applied Biosystems ABI StepOne Plus PCR system (ThermoFisher Scientific, Waltham, MA, USA) with the previously stated TaqMan Assay under the following conditions: one cycle of 48 °C for 30 min, 95 °C for 20 s, and 40 cycles of 95 °C for 3 s and 60 °C for 30 s. Normalization was carried out against 28S rRNA, which was used as a housekeeping gene. To correct for differences in RNA levels between samples within the experiment, the correction factor for each sample was calculated by dividing the mean threshold cycle (Ct) value for the 28S rRNA-specific product for each sample by the overall mean Ct value for the 28S rRNA-specific product from all samples. The corrected cytokine mean was calculated as follows: (average of each replicate × cytokine slope)/(28S slope × 28S correction factor). Fold changes in mRNA levels were calculated from mean 40 Ct values [22,25]. Each sample was run in triplicates for technical replication.

### 2.8. Statistical Analysis for qRT-PCR and Microbiota Analysis

Cytokine mRNA expressions for control and treated ceca from days 2 and 4 were quantitated using a method described by Kaiser et al. [24] and Moody et al. [26]. Statistical analysis was performed with SAS (version 9.4, Cary, NC, USA) of the data collected from each trial for the qRT-PCR data. The Shapiro–Wilk’s test for normality was used to determine if the fold change within each group was parametric or non-parametric, with an alpha of 0.05. For all analyses, statistical significance was considered if *p* ≤ 0.05. All data were found to be non-parametric and were summarized as median values. An ad hoc analysis using the Kruskal–Wallis test was conducted to determine where the statistical differences lie between treatments. The ceca samples for the IL-1β, IL-6, IL-10, TNF-α, and IFN-γ were calculated with the 40-C_t_ method, as outlined by Eldaghayes et al. [22]. The results were reported in fold change values.

The frequencies of bacterial families were submitted to the Kruskal–Wallis test under non-parametric one-way ANOVA, and if there were significant differences seen (*p ≤* 0.05), a post hoc analysis using Dunn’s multiple comparison test to separate the means using GraphPad Prism (version 8.0, La Jolla, CA, USA). For the microbiota correlation analysis, Pearson correlation analysis between the mean percentage of bacterial families and the fold change of cytokines was performed using SAS (version 9.4, Cary, NC, USA) with *α* ≤ 0.05. The Pearson correlation was carried out within each treatment on day 2 and day 4.

## 3. Results

### 3.1. Real-Time Quantitative RT-PCR Results

All results are reported as averaged triplicate experiments and compared to the control (T1). There was a significant upregulation of pro-inflammatory cytokines, specifically IL-1β and IL-6, in challenged birds fed with normal starter formula (T2), at day 2 post-infection (Figure 1A,B). This upregulation was much less marked in challenged birds fed with the subtherapeutic addition of BMD (T4 vs. T2). Furthermore, on day 2, both IL-10 and TNF-α were expressed in response to *Salmonella* challenge (T2 and T4), regardless of BMD addition in the feed (Figure 1C,D). The expression of IFN-γ in infected birds fed with the addition of BMD (T4) was significantly upregulated, in relation to the other treatments (Figure 1E).

By day 4 post-infection, the expression of IL-10, TNF-α, and IFN-γ remained unchanged (Figure 2C–E) in comparison to day 2 post-infection. In contrast to day 2, the profile of pro-inflammatory cytokines IL-1β and IL-6 did not change much in upregulation at day 4 in challenged birds fed with normal starter formula (T2) (Figure 2A,B). However, the same pro-inflammatory cytokines were strongly upregulated at that time in challenged birds fed with subtherapeutic BMD (T4). For birds provided only subtherapeutic BMD (T3), the profile of pro-inflammatory cytokines IL-1β and IL-6 were slightly elevated; the other tested cytokines, IFN-γ, TNF-α, and IL-10, did not yield significant changes in fold change. Finally, the treatment with BMD appeared to invert the inflammatory profile of birds following infection with *S.* Enteritidis at days 2 and 4 post-infection.

Overall, there was significant upregulation of pro-inflammatory and regulatory cytokines in *S.* Enteritidis-challenged birds given subtherapeutic BMD on day 4 than non-infected birds given BMD, indicating a stimulation of the immune system. Notably, there were statistically significant fold changes on day 2 of IL-1β in T2, IL-6 in T4, IL-10 in T2 and T4, TNF-α in T2 and T4, and IFN-γ in T4 (Figure 1). In day 4, there were statistically significant fold changes of IL-1β in T2 and T4, IL-6 in T4, IL-10 in T2 and T4, TNF-α in T2 and T4, and IFN-γ in T4. As previously shown, *Salmonella* induces pro-inflammatory response within 2 days of infection and reprogrammed to anti-inflammatory response by day 4. However, the treatment of birds with bacitracin inhibited the immune alteration in the *Salmonella*-challenged birds (T4) (Figure 2).

### 3.2. Microbiota Composition

To understand whether bacitracin treatment alteration of the cecal microbiota composition could account for the inhibition of immune phenotype in *Salmonella*-challenged birds, the cecal microbiota was evaluated for both days 2 and 4 for all treatment groups. Relative abundances are reported as median values (Table 2), and only the high-frequency bacterial families are reported. The most abundant bacterial families observed in the ceca were *Enterobacteriaceae* and *Clostridiaceae* at day 2 (Figure 3A). There was noticeable increase in bacterial diversity according to age.

On day 2, the 16S rRNA sequence analysis of the cecal microbiota of control birds fed normal starter feed (T1) revealed a microbiota composition dominated by the family *Clostridiaceae*. Birds infected with *Salmonella* and fed normal starter feed (T2) were dominated by *Enterobacteriaceae* (73.6%) followed by lesser presence of *Clostridiaceae* (23.5%). Birds fed bacitracin (T3) had mostly the presence of *Clostridiaceae* (86.5%) followed by a much lesser presence of *Enterobacteriaceae* (4.45%). In the final group with birds challenged with *S.* Enteritidis and provided subtherapeutic BMD in the starter feed (T4), there was mostly the presence of *Enterobacteriaceae* (68.1%) followed by *Clostridiaceae* (25%) in day 2 ceca.

By day 4, there were still mostly *Enterobacteriaceae* and *Clostridiaceae* but with increased presence of *Paenibacillaceae* across all treatment groups. The control group (T1) was still mostly dominated by *Clostridiaceae* (69.1%) but with a marked increase in *Paenibacillaceae* (8.23%) and starting presence of *Bacillaceae* (1.7%). In the treatment group with *S.* Enteritidis challenge fed the control feed (T2), there were mostly *Enterobacteriaceae* (58.9%) followed by *Clostridiaceae* (23.8%). In the non-infected group fed with bacitracin (T3), the cecal composition had varying changes: large *Clostridiaceae* frequency (40.3%) but with an increased frequency of *Lachnospiraceae* (12.3%) and *Paenibacillaceae* (7.25%). The final group of birds challenged with *S.* Enteritidis with subtherapeutic BMD addition (T4) also had increased frequency of *Lachnospiraceae* (1.25%) and *Paenibacillaceae* (1.98%) but was still dominated by *Enterobacteriaceae* (48.5%) and *Clostridiaceae* (34.3%).

The infection by *Salmonella* seemed to provoke an increase in *Enterobacteriaceae* at the detriment of *Clostridiaceae*, regardless of the age of the birds. When comparing the *Salmonella*-challenged group (T2) with the *Salmonella*-challenged with bacitracin in feed group (T4), there is not much difference in diversity and frequency of families on day 2, but there is a notable decrease in *Enterobacteriaceae* and increase in *Lachnospiraceae*, and to a lesser degree, increased frequency of *Clostridiaceae* and decreased frequency of *Paenibacillaceae*. Birds infected with *Salmonella* and fed control feed (T2) were observed to have decreased frequency of *Enterobacteriaceae* (58.9%) as compared to day 2, and a starting presence of *Paenibacillaceae* (3.02%) with not much change in the *Clostridiaceae* frequency (23.8%). The addition of subtherapeutic BMD feed had no major effect on the bacterial composition at day 2 (T3 vs. T1); however, in non-infected birds at day 4, it provoked an increase in *Lachnospiraceae* and other species at the detriment of *Clostridiaceae* (T3 vs. T1). In contrast, *S.* Enteritidis-challenged birds at day 4 with added BMD provoked a significant 10% increase in *Clostridiaceae* compared to a 10% decrease in *Enterobacteriaceae* (T4 vs. T2).

### 3.3. Beta Diversity Index of Cecal Composition

Unifrac was utilized to compare similarities between bacterial communities in the cecal samples across independent time points (day 2 and day 4). Weighted UniFrac plots showed better separation on day 2 than day 4 where more clustering occurred (Figure 4A–F). There were large differences in the weighted UniFrac, but not in the unweighted UniFrac (data not shown) between groups. The ANOSIM of the weighted Unifrac across different ages within each treatment group (Table 3) displays large quantitative differences between T2 and control on day 2 and T4 and control on day 2. However, the differences in microbiota composition between T2 and control (R = 0.62, *p* ≤ 0.001) and T4 (R = 0.59, *p* ≤ 0.001) and control, while still significant, are not as large by day 4, as indicated by lower R values. Overall, while the effect of bacitracin feed was significant both at days 2 and 4, the difference in microbiota composition compared to controls was small, as shown by the small R value between T1 vs. T3 in the ANOSIM analysis.

## 4. Discussion

As sites of persistent colonization of bacteria, the avian ceca are important sites to study and better understand how the poultry GIT microbiota interact with pathogens. Generally, taxonomic diversity increases as the bird ages, enriched mainly with bacteria in the phylum Firmicutes [6]. Immediately after hatch, the chicks are exposed to numerous environmental factors that activate their immune system, leading to low-grade inflammation by increased cytokine and chemokine expression [2]. Previous studies have already observed bacterial families in the ceca, including *Clostridiaceae*, *Lachnospiraceae*, and *Streptococcaceae* as shown in the present results [27]. These results highlight some key bacterial alterations between the gut bacterial communities and a potential link to changes in the intestinal immune function in the crucial first days post-hatch. Therefore, the goal was met to observe the effects of how introducing subtherapeutic BMD during *Salmonella* challenge would affect the phenotype alteration of the ceca.

The addition of subtherapeutic BMD group (T3) affected the pro-inflammatory cytokines IL-1β and IL-6 by the upregulated expression between day 2 and 4, compared to the control (T1). The other cytokine profiles, namely, TNF-α, IL-10, and IFN-γ, were also affected but with a downregulated expression. Meanwhile, the cecal composition comparing day 2 and day 4 showed increasing bacterial diversity, with a marked increase in *Lachnospiraceae* and other species at the detriment of *Clostridiaceae* (T3 vs. T1). This may be due to the subtherapeutic levels provided, in which previous studies [28,29] also did not see significant changes in the cecal community structure but did see changes in the community structure overall. In fact, the addition of pro-inflammatory mediators increases the resistance of the animal to pathogen challenge, specifically a rapid and efficient inflammatory response during pathogen invasion [18].

When an opportunistic pathogen such as *S.* Enteritidis is involved, the results are distinctly different when compared to the control. The pro-inflammatory cytokines IL-1β and IL-6 were significantly upregulated on day 2 and even more upregulated by day 4 for T4, indicating a strong inflammatory response during these two days in the chicks. However, the other cytokines, IL-10, TNF-α, and IFN-γ, were also significantly upregulated on day 2 but then observed to be downregulated overall by day 4. Infections with *S.* Enteritidis and treatment with BMD affected the overall bacterial composition in the ceca, which has also been shown in previous studies with mammalian models as well [30]. Generally, AGPs like BMD improve growth performance through an anti-inflammatory effect in the GI tract [31]. In fact, it is speculated that AGPs may eliminate some members of *Enterobacteriaceae,* but this allows for opportunistic bacteria to proliferate as a result [14]. In this study, the addition of subtherapeutic BMD in the *S.* Enteritidis challenge model (T4) altered the immune system by the induced inflammatory response, compared to the treatment group with just the challenge on starter feed (T2) which was downregulated by day 4. The addition of subtherapeutic BMD in the *S.* Enteritidis-challenged group (T4) also reduced the *Enterobacteriaceae* population by almost 20% from day 2 to day 4. In previous studies from our group [3,32], the phenomenon of phenotypic switch occurred between days 2 and 4 from a pro-inflammatory response to an anti-inflammatory, *Salmonella* disease tolerance response. The inflammatory response, defined by the IL-1β and the IL-6 response, was inhibited on day 2 compared to the *Salmonella*-challenged chicks without BMD treatment. Infected hosts undergo phenotypic shifts in tissue metabolism, metabolic sensors such as mTOR and AMPK, and immune cell functions [33]. The *Salmonella*-challenged treatment (T2) follows the previously documented studies: IL-6 and IL-1β are highly upregulated on day 2 and dramatically reduced by day 4 post-infection [33]. Therefore, it can be speculated that bacitracin is preventing an overstimulation in the earlier stage of *Salmonella* infection. This disruption of the microbiota increases the pathogen’s drive to survive by changing the state of inflammatory response.

According to the UniFrac results, the overall data demonstrated species disappearing or showing up, but with strong differences in abundances of those that were already present. As seen in Figure 4A–F and Table 3, there were large quantitative differences between *S.* Enteritidis-challenged group (T2) and control (T1) on day 2 and *S.* Enteritidis-challenged group with the addition of the subtherapeutic BMD group (T4) and control (T1) on day 2. However, the differences between T2 and control and T4 and control are not as similar by day 4. Juricova et al. [13] concluded that infection with *Salmonella enteritidis* caused delays in the microbiota development of young chicks, which could explain the phenomenon occurring in the present study. This interaction could also affect the immune system of the host, adding another layer of variability that has not been well classified in broilers. Interesting, there were bacterial community differences seen in T2 and T4 compared to the control, but not much between T3 and control. Oakley and Kogut [6] concluded that Proteobacteria, which includes *Salmonella* spp., had strong correlations to IL-6 pro-inflammatory response. We also observed increased IL-10 expression which can be attributed to the gut modulation via bacitracin and young age of the birds [34]. While the presence of these pro-inflammatory cytokines could be attributed to the fluctuations of an unstable gut microbiota environment of young chicks, antibody-mediated immune response has been confirmed due to modulated gut microbiota via antibiotics [6,34].

While increased inflammation may affect the animal’s performance level, appetite and muscle catabolism, the shifting gut microbiota of young broilers would eventually adjust for the imbalance and lead to less inflammation and energy expenditure for the animal as it ages [5,35]. The presence of *Enterobacteriaceae* has been shown to influence the amount of metabolites expressed in the GIT, especially with short-chain fatty acids (SCFAs) [36]. SCFAs are important for energy production for epithelial cells and immune cells including T cells and macrophages [37]. We also observed high frequency of *Enterobacteriaceae*, which *Salmonella enterica* is a part of, on both days in *Salmonella* infected birds treated with subtherapeutic bacitracin (T4). This could be due to bacitracin’s inhibitory action primarily against Gram-positive bacteria even though it is a broad-spectrum antibiotic. Similarly seen in a study by Kumar et al. [38], this may be occurring due to *Salmonella* outcompeting the resident microflora for nutrients, which allows *Enterobacteriaceae* to flourish in the ceca.

The family *Clostridiaceae*, including *Clostridium perfringens*, is of interest in both humans and chickens. On both days tested, the family *Clostridiaceae* persisted, although in varying percentages, since it is generally part of the resident microbiota of the gut [16]. In chickens, pathogenic *C. perfringens* causes necrotic enteritis and is usually prevented by AGPs such as bacitracin [16]. A previous study by Ballou et al. [39] reports increased cecal diversity of bacterial communities, mainly within the order Clostridiales, which our present study confirms. The high frequency of members of the order Clostridiales in the ceca has also been linked to improved growth performance in chickens, which is an important factor for the broiler industry to consider [4,40].

Another minorly present bacterial family was *Lachnospiraceae*, identified as poultry probiotic bacteria, which appears to remain low in abundance until the chickens are much older in age, but this family of bacteria was one of the most modulated members of the cecal community when bacitracin was provided (Figure 3A,B) [4,40]. Depending on the genera, this family has been widely associated for the ability to produce beneficial metabolites, specifically SCFAs, influence dietary digestion, and influence metabolic disease regulation for mammalian hosts as well [41]. This family has been found in mucosal folds of the GI tract, suggesting that their interaction with the lamina propria immune cells makes *Lachnospiraceae,* immune regulators that prevent pathogen colonization [42]. Furthermore, studies have documented an overall reduction of probiotic-related bacteria in the intestinal tract of chickens with the usage of AGPs [4,43,44]. The addition of an SCFA like butyrate has demonstrated to reduce invasion abilities and colonization abilities of *Salmonella* in the host [14,45]. Interestingly, the effects of SCFAs tend to produce anti-inflammatory effects on host immunity [46]. This study has observed the continual upregulation of pro-inflammatory cytokines with chicks provided subtherapeutic BMD (T3) and chicks provided subtherapeutic BMD with *S.* Enteritidis challenge (T4). However, this could be attributed to the unstable microbiota of young chicks or metabolic interactions causing pro-inflammatory effects. The presence of certain SCFAs could explain how bacitracin prevents infection of the cecal epithelium by *Salmonella,* thereby reducing the inflammatory response overall by day 2. Then, by day 4, a response against *Salmonella* is induced at a higher level, as indicated by the large upregulation of pro-inflammatory cytokines in the mRNA expression data. Further studies will need to be conducted to identify what SCFAs are present and how they affect the tissues histologically (lesion scoring or inflammatory evaluation) and immunologically during this crucial time point of chick development.

Overall, our results demonstrate the effects of bacitracin on cecal composition and its interaction with *S.* Enteritidis in young chicks. There is preliminary evidence of a phenotype change in the ceca due to the shifting microbiota with the addition of BMD; the phenotype has been altered by adding subtherapeutic BMD. This is supported by the decrease in *Enterobacteriaceae* and the increase in *Lachnospiraceae* and *Clostridiaceae* from day 2 to day 4. This is further supported by the upregulated inflammatory response of IL-6 and IL-1β on day 2 and day 4 in the *S.* Enteritidis-challenged group provided subtherapeutic BMD. Our group has previously reported the initial occurrence of an immunometabolic reprogramming in the cecal tissue during infection with *S. enterica* [3,32]. The current study validates this phenotype switch event while also providing initial evidence of a phenotype switch in birds provided subtherapeutic BMD and challenged with *S.* Enteritidis. While the study looked at a short time frame, the study demonstrated that the shift in microbiota can affect the immune reaction. These results provide initial insight of the phenotype changes occurring with the cecal bacterial population and the influence on the immune system. Future studies should look at a longer duration period and other segments of the GIT for a wider overview of bacterial communities.

## Figures and Tables

**Figure 1 microorganisms-08-01879-f001:**
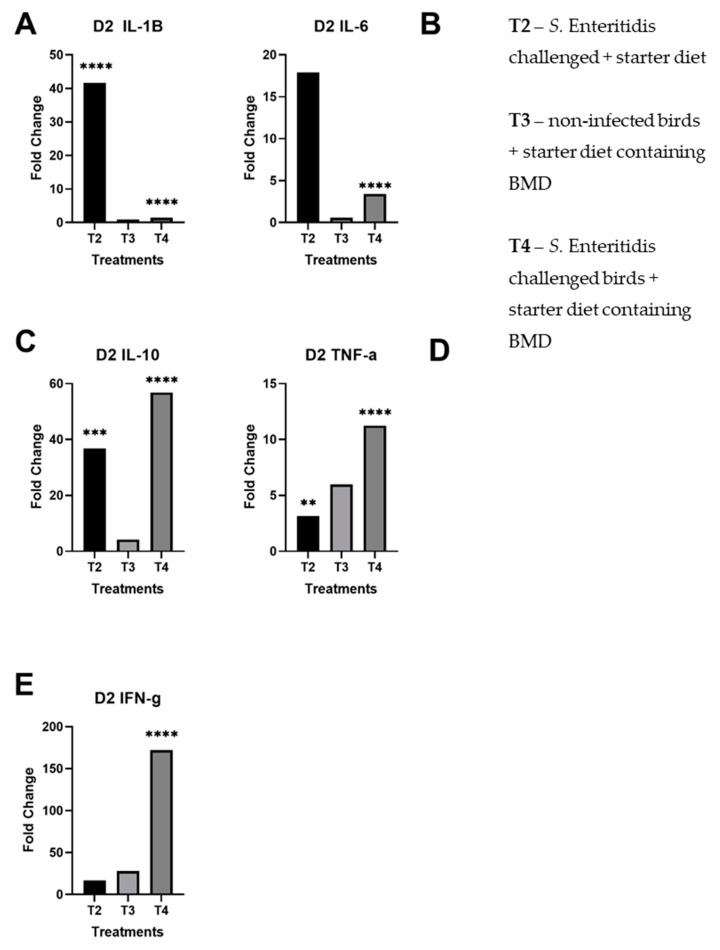
Fold changes of day 2 cecal samples by treatment per panel. These results for each treatment are compared against the control group (T1): (**A**) day 2 fold changes by treatment for IL-1β; (**B**) day 2 fold changes by treatment for IL-6; (**C**) day 2 fold changes by treatment for IL-10; (**D**) day 2 fold changes by treatment for TNF-α; (**E**) day 2 fold changes by treatment for IFN-γ. The starred bars indicate differing levels of significant *p*-values.

**Figure 2 microorganisms-08-01879-f002:**
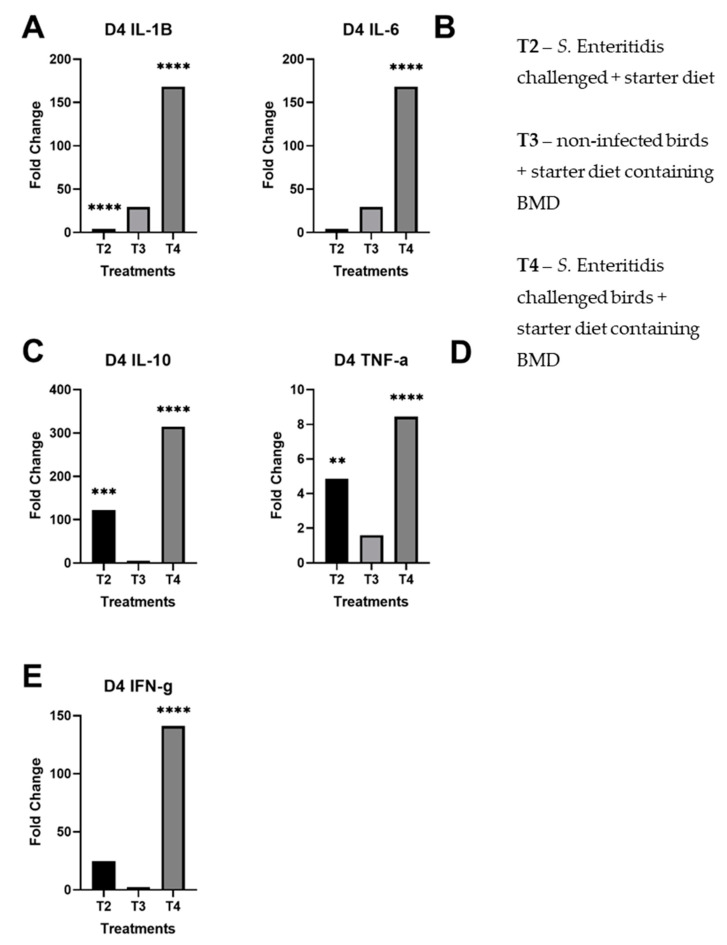
Fold changes of day 4 cecal samples by treatment per panel. These results for each treatment are compared against the control group (T1): (**A**) day 4 fold changes by treatment for IL-1β; (**B**) day 4 fold changes by treatment for IL-6; (**C**) day 4 fold changes by treatment for IL-10; (**D**) day 4 fold changes by treatment for TNF-α; (**E**) day 4 fold changes by treatment for IFN-γ. The starred bars indicate differing levels of significant *p*-values.

**Figure 3 microorganisms-08-01879-f003:**
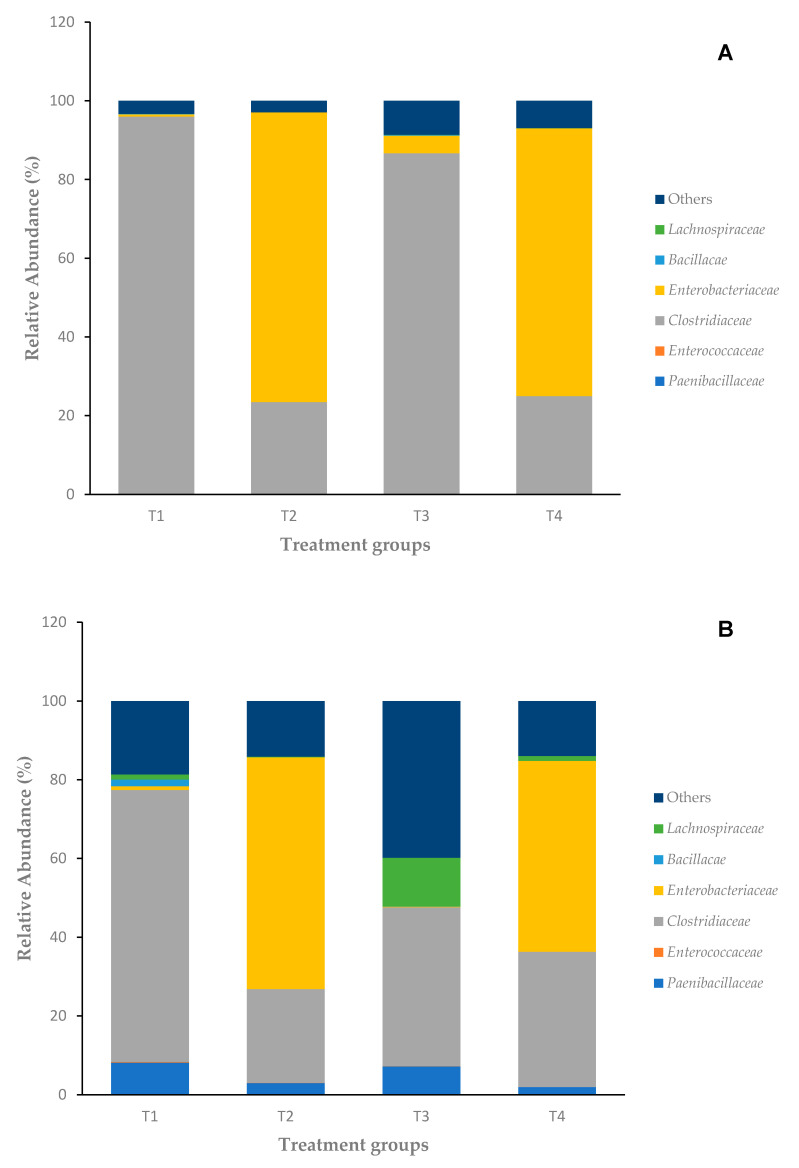
(**A**) Microbiota composition of day 2 chicken ceca; (**B**) microbiota composition of day 4 chicken ceca.

**Figure 4 microorganisms-08-01879-f004:**
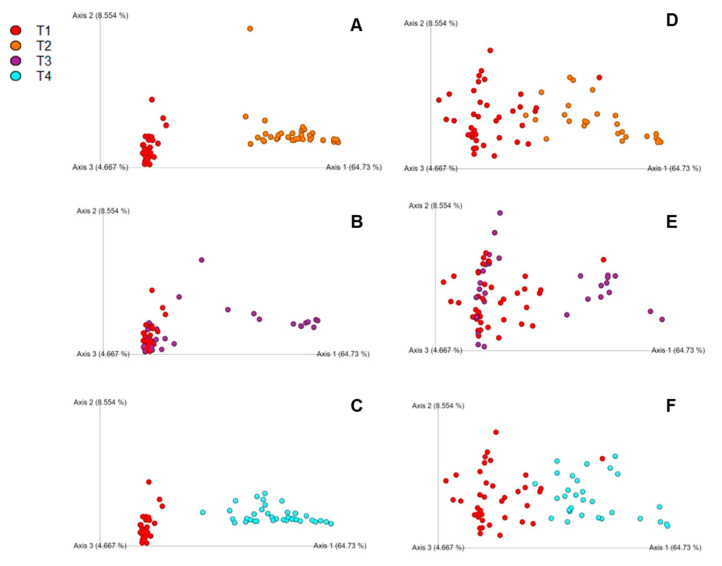
(**A**–**C**) Principle coordinate analysis (PCoA) charts using Unifrac for day 2 comparison of treatments: (**A**) T1 vs. T2, (**B**) T1 vs. T3, (**C**) T1 vs. T4; (**D**–**F**) PCoA charts using Unifrac for day 4 comparison of treatments: (**D**) T1 vs. T2, (**E**) T1 vs. T3, (**F**) T1 vs. T4. The legend is located at the top left of the figure.

**Table 1 microorganisms-08-01879-t001:** Primer and probe sequences.

RNA Target	Probe/Primer Sequences		Accession No.
28S	Probe	5′-(FAM)-AGGACCGCTACGGACCTCCACCA-(TAMRA)-3′	X59733
	F	5′-GGCGAAGCCAGAGGAAACT-3′	
	R	5′-GACGACCGATTGCACGTC-3′	
IL-1β	Probe	5′-(FAM)-CCACACTGCAGCTGGAGGAAGCC-(TAMRA)-3′	AJ245728
	F	5′-GCTCTACATGTCGTGTGTGATGAG-3′	
	R	5′-TGTCGATGTCCCGCATGA-3′	
IL-6	Probe	5′-(FAM)-AGGAGAAATGCCTGACGAAGCTCTCCA-(TAMRA)-3′	AJ250838
	F	5′-GCTCGCCGGCTTCGA-3′	
	R	5′-GGTAGGTCTGAAAGGCGAACAG-3′	
TNF-α	Probe	5′-(FAM)-TGCTGAGAAGGAACAAACTGGTGGT-(TAMRA)-3′	MF000729
	F	5′-CCCATCCCTGGTCCGTAA-3′	
	R	5′- GGCGGCGTATACGAAGTAAAG-3′	
IL-10	Probe	5′-(FAM)-CGACGATGCGGCGCTGTCA-(TAMRA)-3′	AJ621614
	F	5′-CATGCTGCTGGGCCTGAA-3′	
	R	5′-CGTCTCCTTGATCTGCTTGATG-3′	
IFN-γ	Probe	5′-(FAM)-TGGCCAAGCTCCCGATGAACGA-(TAMRA)-3′	Y07922
	F	5′-GTGAAGAAGGTGAAAGATATATCATGGA-3′	
	R	5′-GCTTTGCGCTGGATTCTCA-3′	

**Table 2 microorganisms-08-01879-t002:** The top listed median relative abundance (by %) of observed families in the ceca of day 2 and day 4 birds.

Group	*Paenibacillaceae*		*Lachnospiraceae*		*Clostridiaceae*		*Enterobacteriaceae*		*Bacillaceae*	
	Day 2	Day 4	Day 2	Day 4	Day 2	Day 4	Day 2	Day 4	Day 2	Day 4
T1	0.11 ^a,b^	8.23 ^a^	0	1.32 ^b^	95.9 ^a^	69.1 ^a^	0.55 ^a^	0.94 ^a,c^	0.14 ^a,c^	1.70 ^a^
T2	0 ^b,d^	3.02 ^a,b^	0	0.14 ^b^	23.5 ^b^	23.8 ^b^	73.6 ^b^	58.9 ^b^	0 ^b^	0.02 ^b^
T3	0.18 ^a,c^	7.25 ^a,b^	0.04	12.33 ^a^	86.5 ^c^	40.3 ^b^	4.45 ^c^	0.19 ^c^	0.20 ^c^	0.07 ^b^
T4	0 ^d^	1.98 ^b^	0	1.25 ^b^	25.0 ^b^	34.3 ^b^	68.1 ^b^	48.5 ^b^	0 ^b^	0 ^b^

^a,b,c,d^ The differing superscripts indicate varying significant values (*p* ≤ 0.001).

**Table 3 microorganisms-08-01879-t003:** ANOSIM analysis of weighted Unifrac on different day comparisons within each of the treatment groups.

Age	Comparison	R ^1^	Probability ^2^
Day 2	T1 vs. T2	0.96	**0.001**
	T1 vs. T3	0.1	0.002
	T1 vs. T4	0.98	**0.001**
Day 4	T1 vs. T2	0.62	**0.001**
	T1 vs. T3	0.15	0.002
	T1 vs. T4	0.59	**0.001**

^1^ R is the similarity of comparison: 0 means equally similar, 1 means completely dissimilar. ^2^ Significant differences are set at *p* ≤ 0.001. Significant differences between treatments are bolded.

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
