# Peer review of "A Role for the Microbiota in the Immune Phenotype Alteration Associated with the Induction of Disease Tolerance and Persistent Asymptomatic Infection of Salmonella in the Chicken"

_microorganisms, 2020, doi:10.3390/microorganisms8121879_

Round 1

Reviewer 1 Report

Authors presented very interesting results giving information for further analysis. The manuscript is well written and gives results that could be applicable in the future. There are needed longer experiments, however presented results give some conclusions and can be treated as preliminary study. I have some questions for authors and found some mistakes. Some of my advises could be taken into consideration in the future analysis.

  1. The terms “microbiome” and “microbiota” should not be used as synonyms.
  2. What are other most often Salmonella serovars isolated from chickens in US beside of S. Enteritidis?
  3. The authors should explain in which countries AGPs are allowed and in which are forbidden and why.
  4. Introduction line 42: capital letter should be at the begging of new sentence.
  5. Introduction line 48: GIT - the abbreviation should be expanded as gastrointestinal tract.
  6. Introduction lines 52-54: “Unlike most animals, commercial broiler chicks are placed on recycled litter at hatch, which raises the risk of the naïve gut microbiome of neonatal chicks to be colonized by pathogenic bacteria like Salmonella, and take advantage of this susceptible environment.” Authors should also point other risk factors f.eg. transovarial transmission.
  1. Introduction lines 67-71: Authors should explain why from the broad range of antibiotics, they chose bacitracin methylene disalicylate (BMD) for the study?
  2. In materials and methods authors did not make any control of the eggs. As transovarial transmission is well known phenomenon, authors should check eggs for Salmonella content before starting the experiment. As the eggs could already be infected with Salmonella the enumeration of bacteria could not be precise. Also, the hours and days of experiment could not be precise again due to transovarial – possibility of infection.
  3. Why for Microbiome analysis there were no probes from day 0 (at the time of hatch)?
  4. Line 195: “salmonella” should be changed to “Salmonella”
  5. Figures and Tables: T1-T4 should be explained in captions because it is hard to get back to materials and methods each time.
  6. There should be tables showing results of cfu of Salmonella from Material and methods point 2.3. Bacterial enumeration and detection.

Best regards and successful work in the future.

Author Response

Dear Reviewer,

Thank you so much for taking the time and effort to review my manuscript. I appreciate your helpful feedback and I will respond to each of your comments:

  1. The terms “microbiome” and “microbiota” should not be used as synonyms.
    This is correct. I have fixed this error throughout the manuscript.
  2. What are other most often Salmonella serovars isolated from chickens in US beside of S. Enteritidis?
    We often see serovar Typhimurium but with minor presence of Heidelberg and Newport for human transmission and Gallinarum in poultry specific infections.
  3. The authors should explain in which countries AGPs are allowed and in which are forbidden and why.
    Done
  4. Introduction line 42: capital letter should be at the begging of new sentence.
    Done
  5. Introduction line 48: GIT - the abbreviation should be expanded as gastrointestinal tract.
    Done
  6. Introduction lines 52-54: “Unlike most animals, commercial broiler chicks are placed on recycled litter at hatch, which raises the risk of the naïve gut microbiome of neonatal chicks to be colonized by pathogenic bacteria like Salmonella, and take advantage of this susceptible environment.” Authors should also point other risk factors f.eg. transovarial transmission.
    Done
  1. Introduction lines 67-71: Authors should explain why from the broad range of antibiotics, they chose bacitracin methylene disalicylate (BMD) for the study?
    The other reviewers have mentioned this as well. I have added some support information for this.
  2. In materials and methods authors did not make any control of the eggs. As transovarial transmission is well known phenomenon, authors should check eggs for Salmonella content before starting the experiment. As the eggs could already be infected with Salmonella the enumeration of bacteria could not be precise. Also, the hours and days of experiment could not be precise again due to transovarial – possibility of infection.
    This has been performed but we did not include it in the previous draft. Thank you for bringing it to our attention. The validation has been included.
  3. Why for Microbiome analysis there were no probes from day 0 (at the time of hatch)?
    There are internal controls already built in for baseline purposes.
  4. Line 195: “salmonella” should be changed to “Salmonella”
    Done
  5. Figures and Tables: T1-T4 should be explained in captions because it is hard to get back to materials and methods each time.
    Done
  6. There should be tables showing results of cfu of Salmonella from Material and methods point 2.3. Bacterial enumeration and detection.
    This was an error that was not caught after submission. We ended up not doing enumeration and only did detection. This has been changed in the updated manuscript draft.

Reviewer 2 Report

The manuscript entitled “A role for the microbiota in the immune phenotype alteration associated with the induction of disease tolerance and persistent asymptomatic infection of Salmonella in the chicken”, is reporting a study performed by Annah Lee, Cristiano Bortoluzzi, Rachel Pilla and Michael H. Kogut. Author’s aim was to document the role of the microbiota in the evolution of the cecal immune response to Salmonella, using a model of infection in chicks performed with or without addition of bacitracin at subtherapeutic dose in the diet. In spite of the novelty of the data presented and of the both high scientific interest and socio-economic impact on poultry production sector, the manuscript presents major defects (listed below) concerning the experimental design and the presentation and interpretation of the results that make it unsuitable for publication.

  1. The choice to use bacitracin at subtherapeutic dose to modify the microbiota is not introduced. This is expected to reduce more importantly Gram+ bacterial species in the microbiota as observed with Clostridiaceae (comparison between T3 versus T1 conditions on Figure 3A and 3B). The reduction of Firmicutes and Clostridia associated to pathological inflammatory state of the intestine is well documented (Zechner EL. Inflammatory disease caused by intestinal pathobionts. Curr Opin Microbiol. 2017 Feb;35:64-69. doi: 10.1016/j.mib.2017.01.011. Epub 2017 Feb 10. PMID: 28189956.ref). Therefore, this condition introduces a bias, making the interpretation difficult about the role of the microbiota on the Salmonella-induced host inflammatory response. In addition, the measurement of Salmonella colonization level was not performed after inoculation, leading to important information missing for the interpretation of the results.
  2. The notion of reprogramming as presented at the end of the first paragraph of the results section (3.1) is highly questionable. Firstly, the fold change in immunity genes expression is not explained (by comparison to which condition? probably T1 but this needs to be mentioned). Secondly, according to Figure 1, addition of bacitracin appears firstly to dampen the early inflammatory response (IL1 and IL6) induced by Salmonella infection (T4 versus T2 in Fig 1A and Fig1B). Thirdly, bacitracin appears to delay at day 4 pi this inflammatory response induced by Salmonella (Fig 2A and Fig 2B by comparison to Fig 1A and Fig 1B). Finally, the anti-inflammatory response endorsed only by IL10 (what about TGFb?) appears amplified by bacitracin at day 2 (Fig 1C) and even more importantly at day 4 pi (Fig2C) according to fold changes. Therefore, the description of pro- versus anti-inflammatory response (in fold change of gene expression) according to time post infection must be considered together in the same figure (time course evolution) by merging figures 1 and 2. Moreover, the “inhibition of immune reprogramming in Salmonella-infected birds” appears not appropriate, and should not be used in the manuscript (lines 181-182). The sentence lines 287-289 of the Discussion section is not true. Authors should reconsider the interpretation of their results.
  3. The bioinformatic analysis of microbiota composition raises some important questions.Firstly, the ANOSIM test is described in the Materials and Methods section (lines 133-134) to compare the similarity between bacterial composition within treatments. It appears from Table 3 and Figure 4 that the comparison rather applies between T1 and each other treatment within each time point. Secondly, p≤0.002 is not commonly admitted as no significant differences (threshold at 5% commonly admitted). Thirdly, why there is no comparison of composition between T2 and T4 at each time point, as this appears important regarding the variations in immune profiles, as described in the previous comment (2).

Minor points:

  1. The commercial genetic line of hens, producing fertilized eggs, must be mentioned in the Materials and Methods 2.1 subsection.
  2. The commercial source of bacitracin (BMD) must be mentioned in the Materials and Methods 2.1 subsection
  3. Figures and tables are not easily readable without descriptions of “T1, T2, T3, T4” in the legend.
  4. In the Results subsection 3.2. Microbiome composition, lines 224-226, replace “There was an observed 10.5% decrease in Clostridiaceae between T2 and T4 on Day 4” by “There was an observed 10.5% increase in Clostridiaceae between T2 and T4 on Day 4”.

Author Response

1. The choice to use bacitracin at subtherapeutic dose to modify the microbiota is not introduced. This is expected to reduce more importantly Gram+ bacterial species in the microbiota as observed with Clostridiaceae (comparison between T3 versus T1 conditions on Figure 3A and 3B). The reduction of Firmicutes and Clostridia associated to pathological inflammatory state of the intestine is well documented (Zechner EL. Inflammatory disease caused by intestinal pathobionts. Curr Opin Microbiol. 2017 Feb;35:64-69. doi: 10.1016/j.mib.2017.01.011. Epub 2017 Feb 10. PMID: 28189956.ref). Therefore, this condition introduces a bias, making the interpretation difficult about the role of the microbiota on the Salmonella-induced host inflammatory response. In addition, the measurement of Salmonella colonization level was not performed after inoculation, leading to important information missing for the interpretation of the results.

These are very good points made by the reviewer. I have failed to mention the importance of utilizing BMD; therefore, a section has been added to support this in the introduction. I have also clarified the objective because this seemed to have confused the audience of our main objective, which was to ultimately eliminate as much of the chick gut resident microbiota and then to observe and report the effect when a Salmonella Enteritidis challenge is also included.

2. The notion of reprogramming as presented at the end of the first paragraph of the results section (3.1) is highly questionable. Firstly, the fold change in immunity genes expression is not explained (by comparison to which condition? probably T1 but this needs to be mentioned). Secondly, according to Figure 1, addition of bacitracin appears firstly to dampen the early inflammatory response (IL1 and IL6) induced by Salmonella infection (T4 versus T2 in Fig 1A and Fig1B). Thirdly, bacitracin appears to delay at day 4 pi this inflammatory response induced by Salmonella (Fig 2A and Fig 2B by comparison to Fig 1A and Fig 1B). Finally, the anti-inflammatory response endorsed only by IL10 (what about TGFb?) appears amplified by bacitracin at day 2 (Fig 1C) and even more importantly at day 4 pi (Fig2C) according to fold changes. Therefore, the description of pro- versus anti-inflammatory response (in fold change of gene expression) according to time post infection must be considered together in the same figure (time course evolution) by merging figures 1 and 2. Moreover, the “inhibition of immune reprogramming in Salmonella-infected birds” appears not appropriate, and should not be used in the manuscript (lines 181-182). The sentence lines 287-289 of the Discussion section is not true. Authors should reconsider the interpretation of their results.

I recognize that the term “immune reprogramming” can be controversial because of the weight it carries in the immunology field. I have changed this terminology to reflect a more accurate picture of what we have seen, which would be an “immune phenotype alteration”.

Secondly, I thank the reviewer for noting the error with fold change results, so the explanation has been emphasized that the treatments are all compared to the baseline control (T1). We did not test for TGF-B unfortunately, but this is a good thing to note for future experiments. Lastly, the last few comments made by the reviewer have been noted and changed to reflect better interpretation of the results.

3.The bioinformatic analysis of microbiota composition raises some important questions. Firstly, the ANOSIM test is described in the Materials and Methods section (lines 133-134) to compare the similarity between bacterial composition within treatments. It appears from Table 3 and Figure 4 that the comparison rather applies between T1 and each other treatment within each time point. Secondly, p≤0.002 is not commonly admitted as no significant differences (threshold at 5% commonly admitted). Thirdly, why there is no comparison of composition between T2 and T4 at each time point, as this appears important regarding the variations in immune profiles, as described in the previous comment (2).

The ANOSIM mention in the Materials and Methods has been revised to reflect an accurate description of this analysis. To clarify, T2 and T3 are just internal controls incorporated into the experiment (T2 was to confirm previous data in our lab), not comparing back and forth between T2 and T1. The point of this study was to set the foundation for what happens when subtherapeutic BMD and Salmonella infection would do to the cecal microbiota composition, and therefore, to the immune profile. Furthermore, we used subtherapeutic levels of BMD because we wanted a general reduction in overall bacteria, not total count or largely get rid of everything as with therapeutic levels of BMD.

Reviewer 3 Report

The aim of the article was to prove that cecal microbiota changes may modify the status of Salmonella Enteritidis persistence in chicks. For that purpose, the authors studied the changes in microbiota composition following supplementation of food with bacitracin (antibiotic growth promoter used in broiler breeding). Salmonella Enteritidis was delivered per os at hatch. The authors measured at the same time the mucosa immune response (proinflammatory cytokine gene transcription) at day 2 and day 4 of age in caecum. For both parameters (microbiota composition and mucosal immunity), they compared with/without AGP and with/without Salmonella infection. They observed that the musosal immune transcriptional profile in response to Salmonella infection that would identify a switch from a “resistance” stage to a “tolerance” stage to (as has been previously defined in the same lab) was modified by adding AGP. They conclude that the role of the concomitantly observed changes in cecal microbiota was essential in establishing this resistance versus tolerance status to Salmonella infection in broiler chickens.

Major comment:

Overall the paper is rather confused (i) in introduction to present the chick model of Salmonella infection and to justify the use of the AGP bacitracin and (ii) in discussion that is very difficult to understand.

Validity of the experimental model and choice of analyses:

  • The authors of the present paper used a model of Salmonella infection at hatch that has been previously perfectly characterized in the same lab by Kogut and Arsenault (2017), who proposed that Salmonella infection in the very young chickens can be clearly separated into 3 stages, based on the cecal musosa immune and metabolic responses: 1. Stage 1, qualified as disease resistance with an inflammatory response characterized by expression of pro-inflammatory cytokines and acute heterophilic response in ceca 2 days pi; 2. Stage 2, qualified as disease tolerance with an anti-inflammatory response and an increase of Treg in ceca 4 days pi; 3. Stage 3, return to homeostasis, IL-10 production and persistence of Salmonella. However the authors need to better clarify what they mean by speaking of early infection (4-48h) and late infection (4-14 days) with Salmonella (lines 55-57), what is in the context can be misleading. because indeed resistance to Salmonella infection increases with age. Very young chickens are most susceptible to infection being thus favorite targets to be asymptomatic carriers. Moreover the authors outlined that “chickens present a unique immunological perspective due to the early interaction between the gut immune system and the microbiome” lines 51-52. But this aspect is true for all vertebrates acquiring commensal microbiota at birth. Otherwise explain.

  • We need to know the genetic background of the broiler chickens used. Moreover it is of the utmost importance to know which treatment has been applied to eggs before incubation. In the same line, the authors should clarify about the origin of the microbiota that may colonize the chick gut in the experimental conditions since chickens were maintained in “isolation” units after hatching. If correctly understood, this initial microbiota input should be devoid of any pathogen, especially Salmonella? This microbiota may not reflect the probable higher diversity of gut microbiota colonizing chicks around hatch the field conditions and thus this would narrow the scope of the conclusion by introducing a bias. In addition it sounds somehow strange to speak of number of eggs in the experimental design and not of the number of truly hatched chicks.

  • The authors used bacitracin supplementation which is an AGP which “mechanisms of action on chicken performance is unclear, but supposed to be through microbiota modulation” lines 40-41, in order to modify the microbiota composition of young chickens and to measure the resulting effect on Salmonella persistence. However direct antibiotic effect on gut pathogens and even on microflora is still debated, since this antibiotic is used at sub-therapeutic doses for growth promoting effects. Moreover several non-antibiotics effects of bacitracin have been suspected on gut epithelium and/or immune cells resulting either in pro- or anti-inflammatory effects (Niewold, 2007). Consequently effect of bacitracin on gut microbiota may result from a direct effect on gut microbiota composition and/or an indirect one through immune modulation. Thus possible specific mucosal immune system alteration through the use of bacitracin has to be perfectly explored since they may alter the response to Salmonella before drawing any strong conclusion on the role of microbiota composition in the observed gut immune phenotype. All these hypotheses have to be clearly discussed. On the other hand it remains indeed useful for poultry production to unravel clearly the possible additive effects of bacitracin on pathogen bacterial replication, gut commensal microbiota composition and gut immunity to favor establishment of a carrier state for Salmonella. Unfortunately no result of Salmonella Enteritidis loads in cecal contents and in association with the gut epithelium following infection, as a read-out to define the pre-carrier stage (namely before 4 days of infection) related to bacitracin treatment or not (despite what is indicated in Materials and methods). Moreover deeper analysis of the gut microbiota (phyla to genera/species) would have been beneficial for the paper to give information not only about the structure (phyla), but also on specific membership variation. Indeed previous studies have shown that bacitracin may have less a general effect on all phyla than a greater impact on rare species in chicken gut. And these species (that should be from a chicken origin? See previous) may be known to have a specific and major role in chicken metabolism and immunity. With respect to this aspect, it would have been also interesting to identify the range of metabolites produced by the microbiota in the experimental conditions to have an insight of the ones that may be important to shape the immune response, as it has already be done in literature as an overlook of the potential activity of the microbiota depending on presence/absence of Salmonella and presence/absence bacitracin treatment.

  • Gut immune responses:

What is referred to chicken TNF-alpha is in fact chicken IL-8 considering the accession number AJ009800??

Cecal samplings have been performed for cytokine quantitative PCR analysis and “kinome analysi”s. But no results are available for the latter??

Result expression for immune response is very confused and has to be rewritten to be easily understandable. Taking into account Figure 2, and based on the previous model description of Salmonella infection in chicks, cecal immune response during what is called the “resistance to disease” stage ie day 2 pi, exhibited a pro-inflammatory phenotype (expression of IL-1beta, IL-6, IL-8, IL10) for both groups either bacitracin treated or not-treated, despite clearly much lower in the level of the pro-inflammatory response for the infected bacitracin treated group. By day 4, there was a clear shift of the immune response for both groups. Not-treated infected group exhibiting a shift to the so-called disease tolerance group defined as strong decrease of the pro-inflammatory response as expected (except that here the IL-10 response, characterized as a hallmark of the disease tolerance stage by day 4 pi, was here low). On the contrary bacitracin-treated infected chicks boosted a very high pro-inflammatory response by day 4 pi (IL-1beta, IL-6, IL-8) and IFN-gamma plus IL-10 response. Thus in bacitracin infected group can be observed a delayed, but then higher, response to the pathogen. But it is not possible to know at the present time whether this is due to an inhibition of the shift towards disease tolerance since further timing was not measured, or a mere delay of the immune response or an initial overall inhibition of the cecal pro-inflammatory response by bacitracin that might change the maturation timing of the gut epithelial interface/gut immune system. To test this hypothesis, it would be important to check the immune cell population profiles at the cecal level, at least heterophils, macrophages, T cells including Treg, to better define the cellular immune profile after bacitracin treatment and after Salmonella infection in light of what was already published. Salmonella numeration, either in cecal contents or attached to the epithelium, would have been necessary to know whether this immune response was able to clear salmonella infection. Indeed IL-10 expression (considered as a marker of Treg: NB it would have been better to associate also TGF-beta) was also very high in bacitracin infected group day 2 and day 4 pi. The same was true for IFN-gamma that is known to be involved in Salmonella resistance. In addition histopathology analysis of the cecal mucosa would have been useful to link the level of the inflammatory immune profile with possible epithelial damage. Thus this must be clearly discussed.

  • Gut microbiota profiling:

As already said, a deeper analysis would be appropriate to identify specific genera/species changes that may be of major importance for chicken metabolism/immune modulation even in low number, since overall difference among infected groups were in fact not so strong. However treatment with bacitracin did seem to have an effect compared to non-treated non infected controls. Moreover this analysis is important also to know whether the experimental microbiota was representative of a broiler gut microbiota. Indeed 3 experiments were pooled for analysis (as the number of samples in figure 4 did appear to show. If that was true, it has to be explained why all the putative samples were not included). But explain how the initial microbiota seeding would be identical: see previous comment.

Figure legends should avoid repetition and be fully informative.

Discussion has to be much improved.

Lines 275-279: not understandable.

Line 281: as understood from literature, “Salmonella disease tolerance”?

Lines 285-287: not proved.

Lines 303-304: ?

Line 307: “the presence of these signaling molecules”?

Line 309 : Bacitracin as an antagonist?

Line 316: “fold change” of what?

Lines 318-319: ??

Lines 335-337: ?? Tregs number has not been checked in cecal mucosa.

Line 338-339: why the authors are debating on the effect of hich abundance of enterococcaceae since they observed a low abundance?

Lines 348-350: The authors noted an increase of lachnospiraceae after bacitracin treatment, seemingly contrary to literature. These bacteria produce SCFAs able to reduce invasion and colonization of Salmonella in addition to display an anti-inflammatory effect (cf literature). Thus this can  explanation of what was observed. Thus the conclusion would be that Bacitracin prevents infection of cecal epithelium by Salmonella and reduce the inflammatory response overall at day 2. But later on, the anti-Salmonella response is induced at a higher level. This was not really discussed by the authors.

Line 356: No, data do not demonstrate the interaction of bacitracin with Salmonella.

Lines 356-357: Not proved sensu stricto.

Lines 361-363: Yes perhaps, but not proved (further timing of infection are lacking).

Line 367: why indirect?

Lines 367-369: not understandable.

Lines 369-370: This questions the initial very emphatic conclusions of the present study.

Lines 371-372: OK.

To conclude, the author overstated their conclusion that “the microbiota has a role in the immune phenotype alteration associated with the induction of disease tolerance and persistent asymptomatic infection of Salmonella in the chicken” in the view of present data:

  • Bacitracin treatment has changed the “canonical” immune profiling in cecum at day2 and at day4 post-Salmonella infection of chicks at hatch compared to non-treated infected control, towards a lower pro-inflammatory response at day 2 followed by a mixed higher inflammatory, IL-10 (tolerance) and IFN-gamma (resistance) response at day 4 (cf literature). But we have no information on the following steps of the infection (programming of a true resistant status with elimination of Salmonella or of a delayed tolerant status with asymptomatic carriage)
  • We observed at the same time some alteration of the gut microbiota, seemingly more clear in bacitracin non-infected chicks than in infected chicks. We have no proof of that the microbiota alteration through bacitracin feeding are responsible alone for the changes in the cecal immune responses, since direct effect of bacitracin on the intestinal epithelial barrier and cellular immune response cannot be excluded at the present time.

Author Response

Validity of the experimental model and choice of analyses:

  • The authors of the present paper used a model of Salmonella infection at hatch that has been previously perfectly characterized in the same lab by Kogut and Arsenault (2017), who proposed that Salmonella infection in the very young chickens can be clearly separated into 3 stages, based on the cecal musosa immune and metabolic responses: 1. Stage 1, qualified as disease resistance with an inflammatory response characterized by expression of pro-inflammatory cytokines and acute heterophilic response in ceca 2 days pi; 2. Stage 2, qualified as disease tolerance with an anti-inflammatory response and an increase of Treg in ceca 4 days pi; 3. Stage 3, return to homeostasis, IL-10 production and persistence of Salmonella. However the authors need to better clarify what they mean by speaking of early infection (4-48h) and late infection (4-14 days) with Salmonella (lines 55-57), what is in the context can be misleading. because indeed resistance to Salmonella infection increases with age. Very young chickens are most susceptible to infection being thus favorite targets to be asymptomatic carriers. Moreover the authors outlined that “chickens present a unique immunological perspective due to the early interaction between the gut immune system and the microbiome” lines 51-52. But this aspect is true for all vertebrates acquiring commensal microbiota at birth. Otherwise explain.

 Thank you for pointing this out. I have rephrased this portion of the introduction to make more sense, as I see how confusing this may have been to a reader. For the secondary portion about the “chicks present a unique perspective…”, I have removed this since the reviewer brings up a valid point.

  • We need to know the genetic background of the broiler chickens used. Moreover it is of the utmost importance to know which treatment has been applied to eggs before incubation. In the same line, the authors should clarify about the origin of the microbiota that may colonize the chick gut in the experimental conditions since chickens were maintained in “isolation” units after hatching. If correctly understood, this initial microbiota input should be devoid of any pathogen, especially Salmonella? This microbiota may not reflect the probable higher diversity of gut microbiota colonizing chicks around hatch the field conditions and thus this would narrow the scope of the conclusion by introducing a bias. In addition it sounds somehow strange to speak of number of eggs in the experimental design and not of the number of truly hatched chicks.

 The genetic information of the chicks and the testing done prior to the chicks hatching have been provided, as another reviewer has pointed this out as well. Thank you for catching this!

  • The authors used bacitracin supplementation which is an AGP which “mechanisms of action on chicken performance is unclear, but supposed to be through microbiota modulation” lines 40-41, in order to modify the microbiota composition of young chickens and to measure the resulting effect on Salmonella persistence. However direct antibiotic effect on gut pathogens and even on microflora is still debated, since this antibiotic is used at sub-therapeutic doses for growth promoting effects. Moreover several non-antibiotics effects of bacitracin have been suspected on gut epithelium and/or immune cells resulting either in pro- or anti-inflammatory effects (Niewold, 2007). Consequently effect of bacitracin on gut microbiota may result from a direct effect on gut microbiota composition and/or an indirect one through immune modulation. Thus possible specific mucosal immune system alteration through the use of bacitracin has to be perfectly explored since they may alter the response to Salmonella before drawing any strong conclusion on the role of microbiota composition in the observed gut immune phenotype. All these hypotheses have to be clearly discussed. On the other hand it remains indeed useful for poultry production to unravel clearly the possible additive effects of bacitracin on pathogen bacterial replication, gut commensal microbiota composition and gut immunity to favor establishment of a carrier state for Salmonella. Unfortunately no result of Salmonella Enteritidis loads in cecal contents and in association with the gut epithelium following infection, as a read-out to define the pre-carrier stage (namely before 4 days of infection) related to bacitracin treatment or not (despite what is indicated in Materials and methods). Moreover deeper analysis of the gut microbiota (phyla to genera/species) would have been beneficial for the paper to give information not only about the structure (phyla), but also on specific membership variation. Indeed previous studies have shown that bacitracin may have less a general effect on all phyla than a greater impact on rare species in chicken gut. And these species (that should be from a chicken origin? See previous) may be known to have a specific and major role in chicken metabolism and immunity. With respect to this aspect, it would have been also interesting to identify the range of metabolites produced by the microbiota in the experimental conditions to have an insight of the ones that may be important to shape the immune response, as it has already be done in literature as an overlook of the potential activity of the microbiota depending on presence/absence of Salmonella and presence/absence bacitracin treatment.

These are all good ideas from the reviewer but these will be looked at in our lab soon. We could not possibly perform or look at every option in this manuscript and we hope that this study will provide useful baseline data for future studies.

  • Gut immune responses:

What is referred to chicken TNF-alpha is in fact chicken IL-8 considering the accession number AJ009800??

Thank you for noticing this. I must have input the incorrection accession number.

Cecal samplings have been performed for cytokine quantitative PCR analysis and “kinome analysi”s. But no results are available for the latter??

Thank you for catching that. I have removed any mention of “kinome analysis”

Result expression for immune response is very confused and has to be rewritten to be easily understandable. Taking into account Figure 2, and based on the previous model description of Salmonella infection in chicks, cecal immune response during what is called the “resistance to disease” stage ie day 2 pi, exhibited a pro-inflammatory phenotype (expression of IL-1beta, IL-6, IL-8, IL10) for both groups either bacitracin treated or not-treated, despite clearly much lower in the level of the pro-inflammatory response for the infected bacitracin treated group. By day 4, there was a clear shift of the immune response for both groups. Not-treated infected group exhibiting a shift to the so-called disease tolerance group defined as strong decrease of the pro-inflammatory response as expected (except that here the IL-10 response, characterized as a hallmark of the disease tolerance stage by day 4 pi, was here low). On the contrary bacitracin-treated infected chicks boosted a very high pro-inflammatory response by day 4 pi (IL-1beta, IL-6, IL-8) and IFN-gamma plus IL-10 response. Thus in bacitracin infected group can be observed a delayed, but then higher, response to the pathogen. But it is not possible to know at the present time whether this is due to an inhibition of the shift towards disease tolerance since further timing was not measured, or a mere delay of the immune response or an initial overall inhibition of the cecal pro-inflammatory response by bacitracin that might change the maturation timing of the gut epithelial interface/gut immune system. To test this hypothesis, it would be important to check the immune cell population profiles at the cecal level, at least heterophils, macrophages, T cells including Treg, to better define the cellular immune profile after bacitracin treatment and after Salmonella infection in light of what was already published. Salmonella numeration, either in cecal contents or attached to the epithelium, would have been necessary to know whether this immune response was able to clear salmonella infection. Indeed IL-10 expression (considered as a marker of Treg: NB it would have been better to associate also TGF-beta) was also very high in bacitracin infected group day 2 and day 4 pi. The same was true for IFN-gamma that is known to be involved in Salmonella resistance. In addition histopathology analysis of the cecal mucosa would have been useful to link the level of the inflammatory immune profile with possible epithelial damage. Thus this must be clearly discussed.

These are great ideas that our lab would love to be able to explore in future studies, especially running histopathology and also looking at immune cells.

  • Gut microbiota profiling:

As already said, a deeper analysis would be appropriate to identify specific genera/species changes that may be of major importance for chicken metabolism/immune modulation even in low number, since overall difference among infected groups were in fact not so strong. However treatment with bacitracin did seem to have an effect compared to non-treated non infected controls. Moreover this analysis is important also to know whether the experimental microbiota was representative of a broiler gut microbiota. Indeed 3 experiments were pooled for analysis (as the number of samples in figure 4 did appear to show. If that was true, it has to be explained why all the putative samples were not included). But explain how the initial microbiota seeding would be identical: see previous comment.

 We intend to do a metabolic analysis (specifically with a peptide array) at another time since we felt including both the immune and metabolism would have made it an extensive manuscript. Furthermore, we also intend to identify down to species level in the future. Unfortunately for this study, the analysis was not able to do so accurately.

Figure legends should avoid repetition and be fully informative.

Discussion has to be much improved.

Lines 275-279: not understandable. REWORDED

Line 281: as understood from literature, “Salmonella disease tolerance”? DONE

Lines 285-287: not proved. REWORDED

Lines 303-304: ? REWORDED TO MAKE MORE SENSE

Line 307: “the presence of these signaling molecules”? I’ve realized this sentence had been moved and was out of place. It has been moved to its proper location in the discussion.

Line 309 : Bacitracin as an antagonist? FIXED

Line 316: “fold change” of what? FIXED

Lines 318-319: ?? FIXED

Lines 335-337: ?? Tregs number has not been checked in cecal mucosa. Fair point. We feel it may be best if this section is removed as a result.

Line 338-339: why the authors are debating on the effect of hich abundance of enterococcaceae since they observed a low abundance? REMOVED; Good point

Lines 348-350: The authors noted an increase of lachnospiraceae after bacitracin treatment, seemingly contrary to literature. These bacteria produce SCFAs able to reduce invasion and colonization of Salmonella in addition to display an anti-inflammatory effect (cf literature). Thus this can  explanation of what was observed. Thus the conclusion would be that Bacitracin prevents infection of cecal epithelium by Salmonella and reduce the inflammatory response overall at day 2. But later on, the anti-Salmonella response is induced at a higher level. This was not really discussed by the authors.

Line 356: No, data do not demonstrate the interaction of bacitracin with Salmonella.

Lines 356-357: Not proved sensu stricto.

Lines 361-363: Yes perhaps, but not proved (further timing of infection are lacking). Agreed. However, we have mentioned this limitation already

Line 367: why indirect? You’re right, this sounds unsure so I have removed it

Lines 367-369: not understandable. I have removed this

Lines 369-370: This questions the initial very emphatic conclusions of the present study. Also a good point. Therefore, we decided to remove this sentence.

Lines 371-372: OK.

To conclude, the author overstated their conclusion that “the microbiota has a role in the immune phenotype alteration associated with the induction of disease tolerance and persistent asymptomatic infection of Salmonella in the chicken” in the view of present data:

  • Bacitracin treatment has changed the “canonical” immune profiling in cecum at day2 and at day4 post-Salmonella infection of chicks at hatch compared to non-treated infected control, towards a lower pro-inflammatory response at day 2 followed by a mixed higher inflammatory, IL-10 (tolerance) and IFN-gamma (resistance) response at day 4 (cf literature). But we have no information on the following steps of the infection (programming of a true resistant status with elimination of Salmonella or of a delayed tolerant status with asymptomatic carriage)
  • We observed at the same time some alteration of the gut microbiota, seemingly more clear in bacitracin non-infected chicks than in infected chicks. We have no proof of that the microbiota alteration through bacitracin feeding are responsible alone for the changes in the cecal immune responses, since direct effect of bacitracin on the intestinal epithelial barrier and cellular immune response cannot be excluded at the present time.

We acknowledge the reviewer’s feedback and as a result, we have removed any overstating of the conclusion. However, we would like to state that this is purely a preliminary trial performed and we fully intend to follow up in another manuscript at another time with more detailed experiments and analyses.

Round 2

Reviewer 2 Report

There are still important inaccuracies and confusions in the description of the results as well as in the discussion.

  • The paragraph 3.1 of the Results section has not gained in clarity. It should not start by an overall description of the situation at day 4; followed by the “boring” description of fold changes gene by gene at the two time points in infected animals (T2 and T4). Since authors wish to determine the involvement of the microbiota in the immune phenotype induced by Salmonella infection, authors should reformulate as follows:

There was a significant and strong upregulation of pro-inflammatory cytokines IL-1 and IL-6 in infected birds fed with normal starter formula (T2), at day 2 post-infection (Figure 1A and 1B). Surprisingly, this up-regulation was much less marked in infected birds fed with the subtherapeutic addition of bacitracin (significant difference between T4 and T2? To be mentioned). Furthermore at day 2, both IL-10 and TNF-a were expressed in response to Salmonella infection, whether or not bacitracin was added to food (Figure 1C and 1D); while IFN-g was only expressed in infected birds fed with the addition of bacitracin (Figure 1E). At day 4 post-infection, the Il-10, TNF-a, and IFN-g profiles of expression remained unchanged (Figure 2C, 2D, 2E) by comparison to day 2 post-infection. By contrast to day 2, the pro-inflammatory cytokines IL-1 and IL-6 were not anymore highly upregulated at day 4 in infected birds fed with normal starter formula (Figure 2A and 2B). However, they were strongly upregulated at that time in infected birds fed with the addition of bacitracin. Finally, the bacitracin treatment of birds appeared to invert the inflammatory profile of birds following infection with Salmonella at 2 and 4 days post-infection.

  • In the subsection 3.2 of the results, each part of the Figure 3 (3A and 3B) appears twice in the manuscript (lines 319 to 322). These are not appropriately cited in the text: line 290, “Figure 3a” should be changed by “Figure 3A”; line 297, “Figure 4” should be changed by “Figure 3B”. The sentence lines 288-289 “Across all treatments, there was an increased diversity…” should be removed as a general comment more appropriately mentioned lines 297-299. The group to which the sentence lines 297-299 refers must be indicated. Line 299-301, what do authors mean by “lesser presence of Clostridiaceae in day 2 ceca”, as this sentence refers to day 4? The next sentence lines 301-302 is a non-concordant repetition of a previous one (lines 297-299). In the following sentence (lines 302-304), authors mention the starting presence of Bacillaceae, but why not the one of Lachnospiraceae? Globally, there is a need of comparative analysis according to time, to infection or not, and according to feed addition of bacitracin or not. This could be formulated as follows:

The infection by Salmonella provoked an increase in Enterobacteriaceae at the detriment of Clostridiaceae, whatever the age of the animals. It can be noticed an increase in the diversity of bacterial species according to age. The addition of bacitracin to feed had no major effect at day 2 (T3 versus T1, T4 versusT2); however in non infected birds at day 4 it provoked an increase in Lachnospiraceae and other species at the detriment of Clostridiaceae (T3 versus T1). By contrast in infected birds at day 4, bacitracin added to food provoked a significant 10% increase in Clostridiaceae in parallel with a 10% decrease in Enterobactericeae (T4 versus T2).

  • In the Discussion section, how authors can say that “BMD did not have much effect on S Enteritidis in this present study” (line 481)? The sentence lines 484-486 is not understandable. The next sentence lines 486-487 “This may explain the mRNA expression data where the IL-1b and IL-6 response in day 2 was inhibited due to the high upregulation of these two cytokines” does not make sense. Line 503, what do authors mean by “This interaction”? In the last paragraph, authors mention the phenotype altered by adding bacitracin and summarize this change as a decrease in Enterobacteriaceae and Clostridiaceae and an increase in Lachnospiraceae. I do not agree with this because Clostridiaceae were increased as Lachnospiraceae at a time post infection (day 4) that must be mentioned. Furthermore, this appeared when inflammatory response was amplified, which is not in agreement with a role of these species notably through SCFA production to moderate inflammation. Globally the discussion section is very difficult to follow because not enough well written and intelligible, in relationship with the results obtained.

Author Response

  • The paragraph 3.1 of the Results section has not gained in clarity. It should not start by an overall description of the situation at day 4; followed by the “boring” description of fold changes gene by gene at the two time points in infected animals (T2 and T4). Since authors wish to determine the involvement of the microbiota in the immune phenotype induced by Salmonella infection, authors should reformulate as follows:

There was a significant and strong upregulation of pro-inflammatory cytokines IL-1 and IL-6 in infected birds fed with normal starter formula (T2), at day 2 post-infection (Figure 1A and 1B). Surprisingly, this up-regulation was much less marked in infected birds fed with the subtherapeutic addition of bacitracin (significant difference between T4 and T2? To be mentioned). Furthermore at day 2, both IL-10 and TNF-a were expressed in response to Salmonella infection, whether or not bacitracin was added to food (Figure 1C and 1D); while IFN-g was only expressed in infected birds fed with the addition of bacitracin (Figure 1E). At day 4 post-infection, the Il-10, TNF-a, and IFN-g profiles of expression remained unchanged (Figure 2C, 2D, 2E) by comparison to day 2 post-infection. By contrast to day 2, the pro-inflammatory cytokines IL-1 and IL-6 were not anymore highly upregulated at day 4 in infected birds fed with normal starter formula (Figure 2A and 2B). However, they were strongly upregulated at that time in infected birds fed with the addition of bacitracin. Finally, the bacitracin treatment of birds appeared to invert the inflammatory profile of birds following infection with Salmonella at 2 and 4 days post-infection.

Thank you so much for your thoughtfully written, suggested paragraph! I was astonished by your commitment to help this manuscript, so I thank you profusely for these comments. I have restructured this section to reflect your suggestions.

  • In the subsection 3.2 of the results, each part of the Figure 3 (3A and 3B) appears twice in the manuscript (lines 319 to 322). These are not appropriately cited in the text: line 290, “Figure 3a” should be changed by “Figure 3A”; line 297, “Figure 4” should be changed by “Figure 3B”. The sentence lines 288-289 “Across all treatments, there was an increased diversity…” should be removed as a general comment more appropriately mentioned lines 297-299. The group to which the sentence lines 297-299 refers must be indicated. Line 299-301, what do authors mean by “lesser presence of Clostridiaceae in day 2 ceca”, as this sentence refers to day 4? The next sentence lines 301-302 is a non-concordant repetition of a previous one (lines 297-299). In the following sentence (lines 302-304), authors mention the starting presence of Bacillaceae, but why not the one of Lachnospiraceae? Globally, there is a need of comparative analysis according to time, to infection or not, and according to feed addition of bacitracin or not. This could be formulated as follows:

The infection by Salmonella provoked an increase in Enterobacteriaceae at the detriment of Clostridiaceae, whatever the age of the animals. It can be noticed an increase in the diversity of bacterial species according to age. The addition of bacitracin to feed had no major effect at day 2 (T3 versus T1, T4 versusT2); however in non infected birds at day 4 it provoked an increase in Lachnospiraceae and other species at the detriment of Clostridiaceae (T3 versus T1). By contrast in infected birds at day 4, bacitracin added to food provoked a significant 10% increase in Clostridiaceae in parallel with a 10% decrease in Enterobactericeae (T4 versus T2).

Thank you for noticing those subtle errors. The duplicated Figure 3 was an issue that only affected the PDF version; it was correct in the .docx format. I apologize for that confusion; it was because the PDF form kept all of the revisions from the Word version. I have also fixed the detailed suggestions and I have also taken the thoughtfully written paragraph into consideration as well for the results section 3.2.

  • In the Discussion section, how authors can say that “BMD did not have much effect on S Enteritidis in this present study” (line 481)? The sentence lines 484-486 is not understandable. The next sentence lines 486-487 “This may explain the mRNA expression data where the IL-1b and IL-6 response in day 2 was inhibited due to the high upregulation of these two cytokines” does not make sense. Line 503, what do authors mean by “This interaction”? In the last paragraph, authors mention the phenotype altered by adding bacitracin and summarize this change as a decrease in Enterobacteriaceae and Clostridiaceae and an increase in Lachnospiraceae. I do not agree with this because Clostridiaceae were increased as Lachnospiraceae at a time post infection (day 4) that must be mentioned. Furthermore, this appeared when inflammatory response was amplified, which is not in agreement with a role of these species notably through SCFA production to moderate inflammation. Globally the discussion section is very difficult to follow because not enough well written and intelligible, in relationship with the results obtained.

Each point has been noted and addressed. I clarified the last statement because I realize how confusing it was. Overall, the other reviewers have asked to clarify the discussion further so I hoped I have achieved this during this second round. Thank you for your continued patience

Reviewer 3 Report

The paper has been much improved in V2 and a lot of the reviewer’s remarks have been taken into consideration. However many gray areas persist that need to be clarified.

  1. Abstract: need to be deeply rewritten to avoid too much repetition.

In addition, the conclusion is still biased and somehow overstated starting from the results (lines 5-9). It can be said that Bacitracin addition in food changed the phenotype of salmonella Enteritidis infected chickens by shifting the cecal microbiota composition and the cecal immune response, altering thus the previously characterized immune phenotype reprogramming between day 2 and day 4 of infection. The results suggest that the striking differences in the cecal (gut?) microbiota composition may be involved, but the underlying mechanisms need to be clarified. “due to” line 5 is not proved.

In the conclusion, qualification of “disease resistance” and “disease tolerance” can’t be used since no data are available on Salmonella load and histological lesions. They can be used has to introduce the chick model of infection with S. Enteritidis (first line).

  1. Introduction: has been improved. References have been added.

However lines 100-103, defining the objectives of this study, are totally unclear.

  1. Materials and Methods: have been completed.

Noteworthy, it is indicated that the S. Enteritidis counts have been performed on the cecal samples, but no data are available linked to the paper. Indeed this data would be very important to draw conclusions. This has been previously asked by the reviewer and not clearly answered.

  1. Results:

Immune response results need to be much improved, as well as the figure legends detailed, as was previously asked by the reviewer. For example, indicate what is the control group for the statistics.

Some statistics challenge. In FIG1: IL-1beta T4; IL-6 T2. In FIG 2: T2 and T3 IL1 beta; T2 IL-6. Please check.

  1. Discussion : Discussion needs much improvement, putting forward clearly each hypothesis in order to clarify the logical thread and debating it against literature. Literature data have to be clearly explained to be understandable without going into the published papers apart from checking them.

Discussing first of the effect of bacitracin on the immune response either direct or indirect through alteration (beneficial or not for the chicken) of the gut microbiota here observed, and then discussing the challenging effect of an opportunistic pathogen such as Salmonella Enteritidis considering the changes in the cecal microbiota in the presence of bacitracin, is necessary to clarify the message.

Lines 370-371: ?? There is a notable change of the cecal microbiota composition from day 2 to day 4 after hatching according to the data (table 2), but no clear differences between the immune profile, between control chickens and control chickens treated with bacitracin. Therefore clarify the message Lines 371-373.

Line 481: There are no data available on S. Enteritidis load in caeca stricto sensu in this study despite what is announced in Materials and Methods? Did the authors consider that abundance of Enterobacteriaceae in their results reflects only the Salmonella Enteritidis replication in the intestine?  But this would need a confirmation, otherwise make a statement in the text. Another important point would have been to know how many Salmonella replicate in the cecal mucosa depending of the treatment in order to have an outlook of the efficiency of the immune response on the bacteria replication. This has to be discussed related to the resistance to tolerant status, in addition to the fate of the carrier status of this chickens that can be detected only over the time.

Lines 493-494: “ Bacitracin is preventing overstimulation”. But to characterize plainly the cecal immune profile, the authors should integrate all the data ie TNF-alpha (inflammatory cytokine), IL-10 and notably IFN-gamma that are stimulated after Salmonella infection. This is very important when speaking about “overstimulation”. When the expression of these cytokines is stimulated, this is an immune response. Why the authors have focused their discussion only on IL-1beta and IL-6?

Moreover, as already noted in the previous report, we have no information on the lesions and the afflux of immune cells in the cecal mucosa in the different groups, that would assign precisely the type of immune response to inflammation or not. This has to be clearly acknowledged in the text.

Lines 508-509: That does not help very much with the interpretation. Please explain clearly.

Paragraph “inflammation” lines 510-541, is an example of total lack of clarity. “Antagonist” and “signaling molecules” have to be qualified. As understood from the text, Bacitracin has an overall anti-inflammatory effect by itself, either direct or through alteration of the commensal gut microbiota (increase of Lachnospiraceae). This anti-inflammatory effect would prevent the initial immune response to S. Enteritidis? However Bacitracin would favor the development of Salmonella by favoring the expression of anti-microbial molecules?? And by eliminating the group of Gram+ bacteria that normally compete for nutrients with Salmonella? Thus the final result would be an increase of the number of Salmonella in the ceca that would stimulate an excess of immune response to the pathogen by day 4? But in fact there is rather less abundance of Enterobacteriaceae in ceca of infected chickens treated with Bacitracin compared to infected chickens by day 4? Question: as a result of the immune response (cf IFN-gamma)?. Please re-explain much more clearly and methodically.

References should not be added in conclusion. Line 570: “resulting in the altering downstream energy requirements of the host”, not proved in the present study.

Author Response

The paper has been much improved in V2 and a lot of the reviewer’s remarks have been taken into consideration. However many gray areas persist that need to be clarified.

  1. Abstract: need to be deeply rewritten to avoid too much repetition.

In addition, the conclusion is still biased and somehow overstated starting from the results (lines 5-9). It can be said that Bacitracin addition in food changed the phenotype of salmonella Enteritidis infected chickens by shifting the cecal microbiota composition and the cecal immune response, altering thus the previously characterized immune phenotype reprogramming between day 2 and day 4 of infection. The results suggest that the striking differences in the cecal (gut?) microbiota composition may be involved, but the underlying mechanisms need to be clarified. “due to” line 5 is not proved.

In the conclusion, qualification of “disease resistance” and “disease tolerance” can’t be used since no data are available on Salmonella load and histological lesions. They can be used has to introduce the chick model of infection with S. Enteritidis (first line).

 Well noted. Thank you for this clarification!

  1. Introduction: has been improved. References have been added.

However lines 100-103, defining the objectives of this study, are totally unclear.

 Thank you for noticing. I have clarified this section a bit more.

  1. Materials and Methods: have been completed.

Noteworthy, it is indicated that the S. Enteritidis counts have been performed on the cecal samples, but no data are available linked to the paper. Indeed this data would be very important to draw conclusions. This has been previously asked by the reviewer and not clearly answered.

 Thank you for noticing this detail. I thought I had removed all trace of “counts” since we did not evaluate this in the manuscript. I went over it and removed any mention of colony counts.

  1. Results:

Immune response results need to be much improved, as well as the figure legends detailed, as was previously asked by the reviewer. For example, indicate what is the control group for the statistics.

Some statistics challenge. In FIG1: IL-1beta T4; IL-6 T2. In FIG 2: T2 and T3 IL1 beta; T2 IL-6. Please check.

 I am not sure what happened in the PDF format but the word format had these corrections already done, per your request in the previous round. I apologize if that was not indicated in the format you’ve seen. It will be reflected in the most current draft. In terms of the statistics challenge, these results have been confirmed and verified once again to be correct.

  1. Discussion : Discussion needs much improvement, putting forward clearly each hypothesis in order to clarify the logical thread and debating it against literature. Literature data have to be clearly explained to be understandable without going into the published papers apart from checking them.

Discussing first of the effect of bacitracin on the immune response either direct or indirect through alteration (beneficial or not for the chicken) of the gut microbiota here observed, and then discussing the challenging effect of an opportunistic pathogen such as Salmonella Enteritidis considering the changes in the cecal microbiota in the presence of bacitracin, is necessary to clarify the message.

I have incorporated these edits into the document. Thank you for your detailed feedback!

Lines 370-371: ?? There is a notable change of the cecal microbiota composition from day 2 to day 4 after hatching according to the data (table 2), but no clear differences between the immune profile, between control chickens and control chickens treated with bacitracin. Therefore clarify the message Lines 371-373.

Thank you for catching this. I tried to remove statements that were too bold in the first round.

Line 481: There are no data available on S. Enteritidis load in caeca stricto sensu in this study despite what is announced in Materials and Methods? Did the authors consider that abundance of Enterobacteriaceae in their results reflects only the Salmonella Enteritidis replication in the intestine?  But this would need a confirmation, otherwise make a statement in the text. Another important point would have been to know how many Salmonella replicate in the cecal mucosa depending of the treatment in order to have an outlook of the efficiency of the immune response on the bacteria replication. This has to be discussed related to the resistance to tolerant status, in addition to the fate of the carrier status of this chickens that can be detected only over the time.

This statement has been removed from the manuscript, per this reviewer and at another reviewer’s request. It created too much confusion and the points you’ve made bring up good points to its presence in the text.

Lines 493-494: “ Bacitracin is preventing overstimulation”. But to characterize plainly the cecal immune profile, the authors should integrate all the data ie TNF-alpha (inflammatory cytokine), IL-10 and notably IFN-gamma that are stimulated after Salmonella infection. This is very important when speaking about “overstimulation”. When the expression of these cytokines is stimulated, this is an immune response. Why the authors have focused their discussion only on IL-1beta and IL-6?

I have corrected this by adding more emphasize on the other cytokines in the results and discussion. This was not caught on our end and we apologize for overlooking this fact.

Moreover, as already noted in the previous report, we have no information on the lesions and the afflux of immune cells in the cecal mucosa in the different groups, that would assign precisely the type of immune response to inflammation or not. This has to be clearly acknowledged in the text.

I apologize for not catching this in the previous round. I have added a statement of limitation to address this.

Lines 508-509: That does not help very much with the interpretation. Please explain clearly.

This has been noted and addressed in line.

Paragraph “inflammation” lines 510-541, is an example of total lack of clarity. “Antagonist” and “signaling molecules” have to be qualified. As understood from the text, Bacitracin has an overall anti-inflammatory effect by itself, either direct or through alteration of the commensal gut microbiota (increase of Lachnospiraceae). This anti-inflammatory effect would prevent the initial immune response to S. Enteritidis? However Bacitracin would favor the development of Salmonella by favoring the expression of anti-microbial molecules?? And by eliminating the group of Gram+ bacteria that normally compete for nutrients with Salmonella? Thus the final result would be an increase of the number of Salmonella in the ceca that would stimulate an excess of immune response to the pathogen by day 4? But in fact there is rather less abundance of Enterobacteriaceae in ceca of infected chickens treated with Bacitracin compared to infected chickens by day 4? Question: as a result of the immune response (cf IFN-gamma)?. Please re-explain much more clearly and methodically.

I have removed the statements involving antagonist and signaling molecules since the reviewer is correct; we have neither any experimental data involving this, nor any claim to support it.

References should not be added in conclusion. Line 570: “resulting in the altering downstream energy requirements of the host”, not proved in the present study

Comment noted; removed.

Round 3

Reviewer 2 Report

No comments.

Author Response

Reviewer has not left any comments for the authors to correct.

Reviewer 3 Report

Minor revision :

Line 83-84: not clear

Line 205: “unexpected”? cf per os Salmonella infection model of chicks.

Line 251: at? day 2

Line 337: …and a potential link to changes in intestinal immune function?

Line 355 : As can be supposed from the text, the authors analyzed the effect of Salmonella infection in the context of BMD treatment?  If the authors are speaking of comparison of T4/T3, no statistical comparison has been made. If the authors are speaking of T4/T1, there was no significant reduction of IL-10 and IFN-gamma between day 2 and day 4. Please clarify.

Line 356 : No: infection by salmonella and treatment with BMD ..

Line 369: … day 4, and the inflammatory response, defined by the IL-1beta and the IL-6 response was inhibited on day 2 compared to salmonella infection without BMD treatment.

Author Response

Line 83-84: not clear

I’ve attempted to clarify the sentence but if you could let me know why it’s confusing for you, I’d have a better sense. The authors agree it is clear from our position.

Line 205: “unexpected”? cf per os Salmonella infection model of chicks.

Removed

Line 251: at? day 2

Corrected

Line 337: …and a potential link to changes in intestinal immune function?

Yes, thank you for this improved suggestion!

Line 355 : As can be supposed from the text, the authors analyzed the effect of Salmonella infection in the context of BMD treatment?  If the authors are speaking of comparison of T4/T3, no statistical comparison has been made. If the authors are speaking of T4/T1, there was no significant reduction of IL-10 and IFN-gamma between day 2 and day 4. Please clarify.

I see your concern and I have addressed it to clarify.

Line 356 : No: infection by salmonella and treatment with BMD ..

Corrected

Line 369: … day 4, and the inflammatory response, defined by the IL-1beta and the IL-6 response was inhibited on day 2 compared to salmonella infection without BMD treatment.

Reformatted. Thank you!